# Lanczos spectrum for random operator growth

**Tran Quang Loc**

Department of Theoretical Physics, University of Science, Ho Chi Minh City 70000, Vietnam
Vietnam National University, Ho Chi Minh City 70000, Vietnam

**Abstract**

Krylov methods have reappeared recently, connecting physically sensible notions of complexity with quantum chaos and quantum gravity. In these developments, the Hamiltonian and the Liouvillian are tridiagonalized so that Schrodinger/Heisenberg time evolution is expressed in the Krylov basis. In the context of Schrodinger evolution, this tridiagonalization has been carried out in Random Matrix Theory. We extend these developments to Heisenberg time evolution, describing how the Liouvillian can be tridiagonalized as well until the end of Krylov space. We numerically verify the analytical formulas both for Gaussian and non-Gaussian matrix models.

# 1   Introduction

In [1], the concept of Krylov complexity (K-complexity) was introduced as a novel quantifier of the growth of operators under Heisenberg time evolution. This proposal was extended to Schrodinger time evolution in [2], where it was shown that the spread of the wave function was minimized in the Krylov basis, paving the way to the notion of "spread complexity". In these proposals, the dynamics of states/operators are studied in the Krylov basis, to be reviewed below. In this Krylov basis, the Hamiltonian/Liouvillian takes a tridiagonal form, and the tridiagonal elements are sometimes called the "Lanczos spectrum". The evolution equation is simplified into a one-dimensional chain model, where the hopping parameters are the Lanczos coefficients. The state/operator spreads over larger and larger subspaces of the Hilbert space (potentially reaching the full Hilbert space), as long as the hopping parameters do not vanish.[1]

The analysis of [1] concerned the initial time-lapse of operator evolution, where spread complexity grows exponentially fast. This corresponds to the first edge of the Lanczos spectrum, which grows linearly in this regime. On the other hand, in a finite system, the complexity should saturate at some time scale, typically exponentially large in the entropy of the system. In the context of spread complexity, the origin of this saturation is transparent in a finite system, since then we can only explore so many linearly independent states. More abstractly, as mentioned above, the saturation of spread complexity turns out to correspond to the fact that the Lanczos spectrum (as the Hamiltonian spectrum) is finite, and the off-diagonal Lanczos elements need to vanish at the other edge of the spectrum. Starting with [8], this Lanczos descent has been studied in several places [2, 18, 44, 45, 57, 61].

---

[1]The literature in the topic has grown substantially in the last couple of years, see [3–69].

Since the Lanczos descent controls the saturation of complexity and is intimately related to the finiteness and discreteness of the Hilbert space, it is important to deepen its origin. In particular, it would be desirable to have analytical approaches to such behavior. For Schrodinger (state) evolution, such Lanczos descent was understood on general grounds in [18], where it was verified for Random Matrix Theory. More recently, such formulas have been seen to hold in more general systems, and have been conjectured to discern between chaotic and integrable dynamics [69]. The objective of this article is to extend several aspects of these recent discussions concerning the full Lanczos spectrum of Schrodinger evolution to the original scenario considered in [1], namely that of Heisenberg time evolution. In this vein, we derive the bulk part of the Lanczos spectrum of the Liouvillian superoperator for several instances of Random Matrix Theory, including both Gaussian and non-Gaussian cases.

The article is organized as follows: In Section (2), we introduce the concepts of operator growth and Krylov basis. In Section (3), we review two methods from [18] for analytically deriving the Lanczos spectrum starting from the generic densities of operators. Section (4) explains our method for deriving Liouvillian density from Hamiltonian density. In Section (5), which constitutes the main part of the article, we start by comparing simulation and analytical approaches and then apply our analysis to various models of interest, including GUE, non-Gaussian ensembles, and the Sachdev-Ye-Kitaev (SYK) model [70,71]. We end in Section (6) with some discussion and questions for the future.

## 2    Operator Growth in Krylov basis

In the Heisenberg picture, the time evolution of an operator $\mathcal{O}$ is determined by the equation

$$\partial_t \mathcal{O} \equiv \mathcal{L}\mathcal{O} = -i[H, \mathcal{O}]. \tag{1}$$

Here, the Liouvillian superoperator, denoted as $\mathcal{L} = -i[H, \cdot]$, drives this evolution. The solution to this equation can be formally written as $\mathcal{O}(t) = e^{iHt}\mathcal{O}_0 e^{-iHt} = e^{i\mathcal{L}t}\mathcal{O}_0$, where $\mathcal{O}_0$ is the initial operator. Taylor expanding we can reconfigure this expression as

$$\mathcal{O}(t) = \sum_{n=0}^{\infty} \frac{(it\mathcal{L})^n}{n!}\mathcal{O}_0 = \sum_{n=0}^{\infty} \frac{(it)^n}{n!}\tilde{\mathcal{O}}_n, \tag{2}$$

where we have defined $\tilde{\mathcal{O}}_n \equiv \underbrace{[H, ..[H, \mathcal{O}_0]]}_{n \text{ times}} = \mathcal{L}^n \mathcal{O}_0$ with $n = 0, 1, 2, ..., K-1$. Here, the dimension of the Krylov space, $K$, is constrained such that $1 \leq K \leq D^2 - D + 1$, where $D$ is the dimension of the state's Hilbert space, and $D^2$ corresponds to the dimension of the operator's Hilbert space. The dimension of the Krylov space reaches its maximum when Liouvillian's spectrum is entirely non-degenerate, except for the unavoidably null phases [8].

Understanding the nature of a series of nested commutators $\tilde{\mathcal{O}}_n$ is crucial for solving problems related to time evolution, and ascertaining the growth of operators. To this end, one notices that such a series of operators naturally generate an orthonormal basis for the operator algebra. This is known as the Krylov basis. To find such a basis we first need to turn the operator space into a Hilbert space. This is achieved by defining an inner product in the operator algebra. The notion of inner product is not unique as pointed out in [1,6]. For finite-temperature $T = 1/\beta$ (set $k_B = 1$), one can fix the family of inner products as [72],

$$(\mathcal{O}_a | \mathcal{O}_b) = \frac{1}{\beta} \int_0^{\beta} g(\lambda) \langle e^{\lambda H} \mathcal{O}_a^{\dagger} e^{-\lambda H} \mathcal{O}_b \rangle_{\beta} d\lambda, \tag{3}$$

where the thermal expectation value and partition function reads

$$\langle \mathcal{O}_a \rangle_\beta = \frac{1}{Z} \text{Tr}\left(e^{-\beta H} \mathcal{O}_a\right), \qquad Z = \text{Tr}\left(e^{-\beta H}\right), \tag{4}$$

and $g(\lambda)$ is some function on thermal circle $[0, \beta]$ satisfying

$$g(\lambda) > 0, \qquad g(\beta - \lambda) = g(\lambda), \qquad \frac{1}{\lambda} \int_0^\beta d\lambda g(\lambda) = 1. \tag{5}$$

The interesting aspect of these inner products is that they are also applicable in systems with infinite degrees of freedom such as QFT. Nonetheless, in this paper, we stick to the standard convention in finite systems, given by the trace

$$(\mathcal{O}_a | \mathcal{O}_b) = \frac{\text{Tr}\left(\mathcal{O}_a^\dagger \mathcal{O}_b\right)}{\text{Tr}(\mathbb{1})} = \frac{\text{Tr}\left(\mathcal{O}_a^\dagger \mathcal{O}_b\right)}{D}, \tag{6}$$

corresponding to infinite temperature in the previous formulas and assigning a standard operator norm $||\mathcal{O}|| := (\mathcal{O}|\mathcal{O})^{\frac{1}{2}}$. Given such an inner product, the Lanczos algorithm [72, 73], applied to the series of nested commutators, produces an orthonormal basis. In detail, starting from $|\mathcal{O}_n) = \mathcal{L}^n |\mathcal{O})$, we apply the Gram-Schmidt procedure to produce a sequence of Lanczos coefficient $\{b_n\}$, and the set of orthonormal Krylov basis $\{|\mathcal{O}_n)\}$ as

$$|\mathcal{O}_1) = b_1^{-1} \mathcal{L} |\mathcal{O}_0), \tag{7}$$

$$|A_{n+1}) = \mathcal{L} |\mathcal{O}_n) - b_n |\mathcal{O}_{n-1}), \qquad |\mathcal{O}_n) = b_n^{-1} |A_n), \qquad b_n = (A_n | A_n)^{1/2}. \tag{8}$$

The iteration is terminated when the Liouvillian no longer produces a linearly independent state. The Liouvillian superoperator takes a particularly simple form in the Krylov basis, where it becomes a tridiagonal matrix with entries made of the "Lanczos coefficients" $b_n$,

$$\mathcal{L}_{ab} = (\mathcal{O}_a | \mathcal{L} | \mathcal{O}_b) = \begin{pmatrix} 0 & b_1 & & & \\ b_1 & 0 & b_2 & & \\ & \ddots & \ddots & \ddots & \\ & & b_{K-2} & 0 & b_{K-1} \\ & & & b_{K-1} & 0 \end{pmatrix}. \tag{9}$$

In finite-dimensional systems, the tridiagonal matrix within the Krylov basis is recognized as the "Hessenberg form" of the Liouvillian. The full set of Lanczos coefficients will be dubbed the Lanczos spectrum in what follows.

Although we will not need this below, we end this review part by noticing that the Lanczos spectrum can be derived as well from the auto-correlation function $C(t)$, namely the survival amplitude of the initial state,

$$C(t) \equiv (\mathcal{O}_0 | e^{i\mathcal{L}t} | \mathcal{O}_0) = (\mathcal{O}_0 | \mathcal{O}(t)). \tag{10}$$

This is explained in full generality in [1, 2]. We also notice that once we have derived the Krylov basis, the time-dependent operator can be expanded in such a basis as

$$|\mathcal{O}(t)) = \sum_n i^n \phi_n(t) |\mathcal{O}_n), \tag{11}$$

where $C(t) = \phi_0(t)$ and where unitarity implies $\sum_n |\phi_n(t)|^2 = 1$. In the Krylov basis, the Heisenberg equation reads

$$\partial_t \phi_n(t) = b_n \phi_{n-1}(t) - b_{n+1} \phi_{n+1}(t) \,, \tag{12}$$

where $\phi_{-1}(t) = 0$, and $\phi_n(0) = \delta_{n,0}$. In this basis, one concludes that time evolution simplifies to a one-dimensional motion governed by the previous tridiagonal Liouvillian, whose off-diagonal elements are the hoping parameters.

# 3 Lanczos spectrum from Liouvillian spectrum

As discussed above, the main character in the Krylov story is the Lanczos spectrum of the Liouvillian superoperator, which follows the Lanczos algorithm. While finding the exact spectrum in analytical terms is almost an impossible task for interacting theories (akin to finding the exact Hamiltonian spectrum), we can hope to obtain statistical information in the thermodynamic limit assuming thermodynamic quantities such as the density of states. To show this can indeed be achieved, in this section, we take a first small step. We review the technique described in [18] for the state (Hamiltonian) analysis and notice its domain of applicability can be extended to operators, as long as we know Liouvillian density $\rho_{\mathcal{L}}(i\Delta E)$. We will argue for this in two different ways.

## 3.1 Liouvillian block approximation

Focusing on systems with a large entropy (thermodynamic limit), the fundamental assumption (to be numerically verified later) is the existence of a continuous limit for the Lanczos coefficients $b_n$, rewritten in continuous form as $b(x)$ where $x = n/K$. In these scenarios, the strategy involves partitioning the 1-D Krylov chain into $\sqrt{K}$ segments, each with a length of $L = \sqrt{K}$. At the segment boundaries, we set $b_n$ to zero, and within each segment, we assign the remaining $b_n$ values as the average across that segment. [18] then assumes that the impact of these two steps becomes negligible in the thermodynamic limit, under the condition that Lanczos spectrum exhibits sufficiently slow variations within each segment (as $L/K \xrightarrow{N\to\infty} 0$). This approximation has been numerically validated for several cases in [18, 69]. Below we will also verify the operator scenario. With this block approximation, each of the segments becomes a symmetric tridiagonal Toeplitz matrix, with vanishing diagonal elements, of size $\sqrt{K}$. Their eigenvalues are given by

$$E_k = 2b \cos \frac{k\pi}{L+1} \,, \qquad \text{with } k = 1, .., L \,, \tag{13}$$

with real positive $b$. From that, we derive the density of each segment to be

$$\rho_b(E) = \frac{1}{L|dE_k/dk|} = \mathrm{Re}\left[\frac{1}{\pi\sqrt{4b^2 - E^2}}\right] = \frac{H(4b^2 - E^2)}{\pi\sqrt{4b^2 - E^2}} \,, \tag{14}$$

where $H(x)$ is Heaviside function. The $\frac{1}{L}$ factor serves as a normalization constant for integration over the density. In the thermodynamic limit, we conclude that the total density tends to

$$\rho(E) \sim \frac{1}{N} \sum_{n=1}^{S} \frac{L H(4b(nL)^2 - E^2)}{\pi\sqrt{4b(nL)^2 - E^2}} \sim \int_0^1 dx \frac{H(4b(x)^2 - E^2)}{\pi\sqrt{4b(x)^2 - E^2}} \,. \tag{15}$$

This equation relates the continuum approximation of the Lanczos spectrum to the continuous approximation of the density of states. We conclude this formula then applies to the operator scenario if we input in the left-hand side the Liouvillian density.

## 3.2  Liouvillian moment

An ultimately equivalent approach is to consider the Liouvillian in the thermodynamic limit to be well approximated around a given coefficient by an infinite Toeplitz matrix $T(0, b)$. In considering the Liouvillian's tridiagonal nature, the $i, j^{\text{th}}$ entry of $\mathcal{L}^n$, denoted as $[\mathcal{L}^n]_{ij}$, is an $n$-th order polynomial of certain $b_j$ with $|j - i| < n$. Let $k$ be an index within $n$ of $i, j$ and allow $n$ to scale sub-linearly in $K$ as $K$ grows large, such that the difference $|i - k|/K < n/K \to 0$. Assuming that $b(x)$ transition smoothly to their large $K$ limits, we can approximate all instances of $b_i$ by $b_k$ and we obtain

$$[\mathcal{L}^n]_{ij} \sim [T(0, b_k)^n]_{ij} \,. \tag{16}$$

Following [18], for $b = 1$ this can be computed by enumeration of Dyck paths,

$$[T(0, 1)^n]_{ij} = \binom{n}{(n + (i - j))/2} \,. \tag{17}$$

Using the following integral identity

$$[T(0, 1)^n]_{ii} = \binom{n}{n/2} = \int_{-2}^{2} dx \frac{x^n}{\pi\sqrt{4 - x^2}} \,, \tag{18}$$

and that $T(0, b)$ is related to $T(0, 1)$ by a scaling $T(0, b) = bT(0, 1)$, we find

$$\operatorname{tr} \mathcal{L}^n = \sum_i [\mathcal{L}^n]_{ii} \sim \sum_i [T(0, b_i)^n]_{ii} = \int_{-2}^{2} dx \frac{(bx)^n}{\pi\sqrt{4 - x^2}} = \int_{-2b_i}^{2b_i} dE \frac{E^n}{\pi\sqrt{4b_i^2 - E^2}} \,, \tag{19}$$

where we have substitute $E = bx$. Using these expressions, the moments of Liouvillian then read

$$\int dE E^n \rho(E) \sim \frac{1}{K} \sum_i \int_{-2b_i}^{2b_i} dE \frac{E^n}{\pi\sqrt{4b_i^2 - E^2}} \sim \int_0^1 dx \int_{-2b(x)}^{2b(x)} dE \frac{E^n}{\pi\sqrt{4b(x)^2 - E^2}} \,, \tag{20}$$

from which we derive the same Eq. (15). The Lanczos coefficients for operators can be determined from the integral equation by employing deconvolution or Algorithm (1) for states as detailed in [18]. The methodology is outlined in Appendix (A).

## 4  Liouvillian density from Hamiltonian density

We have seen how to derive the Lanczos spectrum of the Liouvillian from the Liouvillian density of states. We now need to find this quantity. Consider a Hilbert space of dimension $N$. Operators are characterized by a dimension of $N^2$. Here, the Liouvillian superoperator can be represented as an $N^2 \times N^2$ matrix. This matrix operates on vectors containing $N^2$ components assembled from the elements of operators. The components of the Liouvillian can be expressed in terms of the original Hamiltonian through the relation (1) as follows,

$$\partial_t \mathcal{O}_{ij} = \sum_{k=1}^{N} \sum_{l=1}^{N} \mathcal{L}_{klij} \mathcal{O}_{kl} = -i \sum_{k=1}^{N} \sum_{l=1}^{N} (H_{ki}\delta_{lj} - \delta_{ki}H_{lj}) \mathcal{O}_{kl} \,, \tag{21}$$

treating the pairs $(k, l), (i, j)$, each of dimension $N^2$, as single indexes comprising two N-component entries. We now assume the Hermitian Hamiltonian has $N$ distinct real eigenvalues $\{E_i\}$ with $(i = 1, ..., N)$. From Eq. (21), the Liouvillian has $\mathcal{L}$ has $N^2$ pure imaginary eigenvalues of the form

$$i\Delta E_{ij} := -i(E_i - E_j),\tag{22}$$

where $i, j = 1, ..., N$. Obviously, $N$ of these are degenerate as $i = j$. Hence, its density comprises the sum of degenerate and non-degenerate parts, encoded in $\delta_{0,\Delta E}$ and $\rho_{\mathcal{L}, \text{ non-deg.}}(i\Delta E)$ respectively as in the following equation,

$$\rho_{\mathcal{L}}(i\Delta E) = \frac{\delta_{0,\Delta E}}{N} + \left(\frac{N-1}{N}\right)\rho_{\mathcal{L}, \text{ non-deg.}}(i\Delta E).\tag{23}$$

The degenerate contribution of $\delta_{0,\Delta E}$ is suppressed by $\frac{1}{N}$. In the thermodynamic limit, as $N$ grows increasingly large, this contribution diminishes. Consequently, the distribution gradually transforms into a smoother form resembling the non-degenerate part $\rho_{\mathcal{L}, \text{ non-deg.}}(i\Delta E)$. It is worth mentioning that the probability is evenly distributed among the N distinct values of $E_i$ (and $E_j$). Therefore, in this limit the non-degenerate spectrum of the Liouvillian constructed from Eq. (21) approximates the convolution of two independent (non-correlated) Hamiltonian densities $\rho_H(E)$,

$$\rho_{\mathcal{L}, \text{ non-deg.}}(i\Delta E) = \text{Re}\left[\int_{E_{\min}}^{E_{\max}} \rho_H(x)\rho_H(\Delta E - x)\mathrm{d}x\right] H(E_{\max} - E_{\min} - |\Delta E|).\tag{24}$$

Regarding the support of the above integral, we examine the real spectrum $\rho_H(E)$ of the Hamiltonian $H$, extending along the real line from $E_{\min}$ to $E_{\max}$. Consequently, the support of the non-degenerate Liouvillian spectrum, $\rho_{\mathcal{L},non-deg.}(i\Delta E)$, is located on the imaginary line $i[E_{\min} - E_{\max}, E_{\max} - E_{\min}]$. In some ranges of $x$ (or $E_i$) and $\Delta E$, the term $(\Delta E - x)$ may fall outside the support $[E_{\min}, E_{\max}]$ of $\rho_H$, rendering the convolution integral referred to in Eq. (24) complex. To render the convolution valid, we can simply disregard the contribution from those ranges by exclusively considering the real part of the convolution integral within its established support. Alternatively, redefining the dummy variable and its integration range, as demonstrated later in Eq. (32), presents another viable approach.

## 5 Lanczos spectrum for random operator growth

In previous sections, we have seen how to derive the continuous limit of the Lanczos spectrum of the Liouvillian just from knowledge of the thermodynamic limit of the density of states of the Hamiltonian. In this section, we apply these formulae in specific examples and verify the approach numerically. Since both analytical and numerical approaches turned out to be involved we start with a brief summary of the specific strategy in each case.

### 5.1 Numerical versus analytical computation

Our numerical computation starts by generating instances of random matrices, followed by finding their eigenvalue spectra. From that, we derive the Liouvillian spectrum, employing the expression described above, where the eigenvalues are given by $\{i(E_i - E_j)\}$. To derive the Lanczos coefficients, we first get the full form of the Liouvillian matrix from the Hamiltonian matrix, directly applying Eq. (21). The spectrum of the Liouvillian can also be obtained from the Liouvillian matrix, serving as a cross-verification for the aforementioned method. Finally, by making use of Hessenberg decomposition and performing a finite number of steps involving Householder's transformation for unitary

similarity transforms, we achieve the tridiagonal Hessenberg representation of the Liouvillian. From this representation, we extract the numerical Lanczos coefficients.

To extend our analysis from GUE to non-Gaussian matrices, the Liouvillian can be reconstructed by combining the GUE eigenvectors matrix with the diagonally stretched eigenvalues mentioned in Section (5.3). The chart for simulation approaches is as follows,

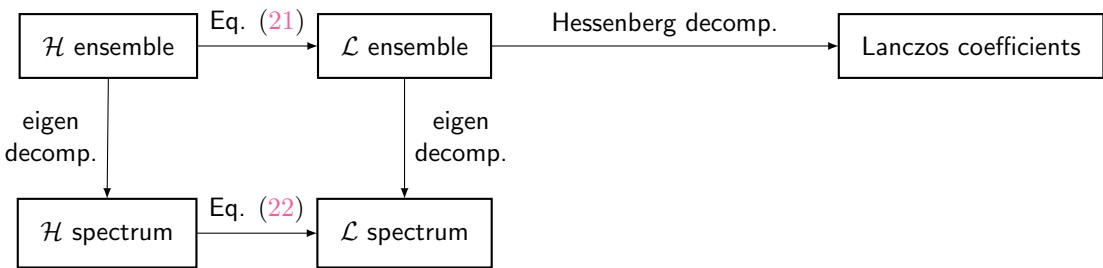

Concerning the analytical computation, we start with an analytical form of a certain Hamilton density, then we employ convolution techniques to derive the corresponding Liouvillian density in the thermodynamic limit. Subsequently, we apply either a deconvolution method or Algorithm (1) to extract the Lanczos coefficients from the obtained Liouvillian density. The chart for the analytic approach is as follows,

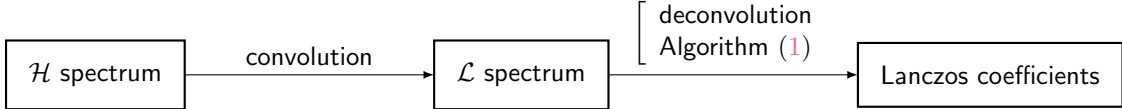

## 5.2 Gaussian Unitary Ensemble

We start with the simplest example, that of the Gaussian unitary ensemble. Gaussian ensembles of random matrices are characterized by Gaussian measure with densities

$$\frac{1}{Z_\beta(N)} \exp\left(-\frac{\beta N}{4} \operatorname{Tr}(H^2)\right), \tag{25}$$

where the Dyson index $\beta = 1, 2, 4$ corresponds to orthogonal, unitary, and symplectic ensembles respectively, and $Z_\beta(N)$ denotes the partition function that normalizes the measure to unity. We consider the Gaussian Unitary Ensemble (GUE) measure with $\beta = 2$, namely

$$\frac{1}{Z_{\mathrm{GUE}}(N)} \exp\left(-\frac{N}{2} \operatorname{Tr}(H^2)\right), \tag{26}$$

where

$$Z_{\mathrm{GUE}}(N) = 2^{\frac{N}{2}} \left(\frac{\pi}{N}\right)^{\frac{N^2}{2}}. \tag{27}$$

The variance in each matrix entry is given by $\sigma^2 = 1/N$. Given that H is hermitian, it possesses $N$ distinct real eigenvalues denoted as $E_i$ for $i = 1, ..., N$. In the large-$N$ limit, the spectrum converges to the Wigner semi-circle distribution

$$\rho_H(x) = \frac{\sqrt{4 - x^2}}{2\pi}. \tag{28}$$

The non-degenerate part of Liouvillian density can be obtained from Eq. (24) as

$$\rho_{\mathcal{L},\,\text{non-deg.}}(i\Delta E) = \text{Re}\left[\int_{-2}^{2}\frac{\sqrt{4-x^2}}{2\pi}\frac{\sqrt{4-(x-\Delta E)^2}}{2\pi}dx\right]H(4-|\Delta E|)\,. \tag{29}$$

The real support of $\rho(x)$ (or $\rho(E_i)$) spans the interval [-2, 2]. Consequently, the Liouvillian also exhibits support within the range of [-4, 4]. We restrict our consideration to the real part of the convolution within this support region. To facilitate the analytical derivation of the integral, we can redefine the integration variable and bounds. Implementing the "Re" function is equivalent to adjusting the integral bounds from [-2, 2] to $[\max(-2, \Delta E - 2), \min(2, \Delta E + 2)]$. Notably, the choice of these bounds depends solely on the sign of $\Delta E$, whether it is positive or negative.

Furthermore, it is worth noting that for every $\Delta E_{ij} = E_i - E_j$ with $i \neq j$, there is a corresponding $\Delta E_{ji} = E_j - E_i$. In other words, the distributions for positive and negative $\Delta E$ are identical, making the Liouvillian density symmetric under the change of sign: $\rho(\Delta E) = \rho(-\Delta E)$. Therefore, it suffices to focus on one sign, opting for positive $\Delta E$, which simplifies our convolution to

$$\rho_{\mathcal{L}\,\text{non-deg.}}(\Delta E > 0) = \text{Re}\left[\int_{-2}^{2}\frac{\sqrt{4-x^2}}{2\pi}\frac{\sqrt{4-(x-\Delta E)^2}}{2\pi}dx\right]\Bigg|_{\Delta E > 0} \tag{30}$$

$$= \int_{\Delta E-2}^{2}\frac{\sqrt{4-x^2}}{2\pi}\frac{\sqrt{4-(x-\Delta E)^2}}{2\pi}dx \tag{31}$$

$$= \int_{\Delta E/4-1}^{-\Delta E/4+1}\frac{2}{\pi^2}\sqrt{x^2-\left(1+\frac{\Delta E}{4}\right)^2}\sqrt{x^2-\left(1+\frac{\Delta E}{4}\right)^2}\,dx \tag{32}$$

$$= \frac{8\left(1-\frac{\Delta E}{4}\right)}{3\pi^2}\left(\left(\frac{\Delta E^2}{16}+1\right)E\left(\sin^{-1}\left(\frac{4-\Delta E}{\Delta E+4}\right)\left|\frac{(\Delta E+4)^2}{(4-\Delta E)^2}\right.\right)\right.$$

$$\left.+\frac{1}{2}\Delta E F\left(\sin^{-1}\left(\frac{4-\Delta E}{\Delta E+4}\right)\left|\frac{(\Delta E+4)^2}{(4-\Delta E)^2}\right.\right)\right)\,, \tag{33}$$

with $F(\phi|k^2), E(\phi|k^2)$ are first and second kind elliptic integral respectively,

$$F\left(\sin^{-1}\left(\frac{4-\Delta E}{\Delta E+4}\right)\left|\frac{(\Delta E+4)^2}{(4-\Delta E)^2}\right.\right) = \int_{0}^{\sin^{-1}\left(\frac{4-\Delta E}{4+\Delta E}\right)}\frac{1}{\sqrt{1-(\frac{4+\Delta E}{4-\Delta E})^2\sin^2\theta}}\,d\theta\,, \tag{34}$$

$$E\left(\sin^{-1}\left(\frac{4-\Delta E}{\Delta E+4}\right)\left|\frac{(\Delta E+4)^2}{(4-\Delta E)^2}\right.\right) = \int_{0}^{\sin^{-1}\left(\frac{4-\Delta E}{4+\Delta E}\right)}\sqrt{1-\left(\frac{4+\Delta E}{4-\Delta E}\right)^2\sin^2\theta}\,d\theta\,. \tag{35}$$

In Fig. (1), we depict the GUE Liouvillian spectrum, presenting a comparison between the analytically derived Liouvillian spectrum obtained by convolving two GUE Hamiltonian spectra (formula (33)), and Liouvillian spectrum numerical simulations. These simulations are conducted for both small $N = 60$ and large $N = 600$, spanning 50 Hamiltonian ensembles. In the simulation, the contribution of degeneracy forms a Dirac distribution of data points, which is attenuated by a factor of $N$, as expected. We find a perfect match.

The Lanczos spectra are obtained analytically using the techniques outlined in Section (3) (whose input is the verified Liouvillian density) and are then compared with the numerical simulations obtained through the procedure described above. The results are illustrated in Fig. (2). Again, we find a perfect match between the analytic and numerical approaches, and thus properly derive the Lanczos descent.

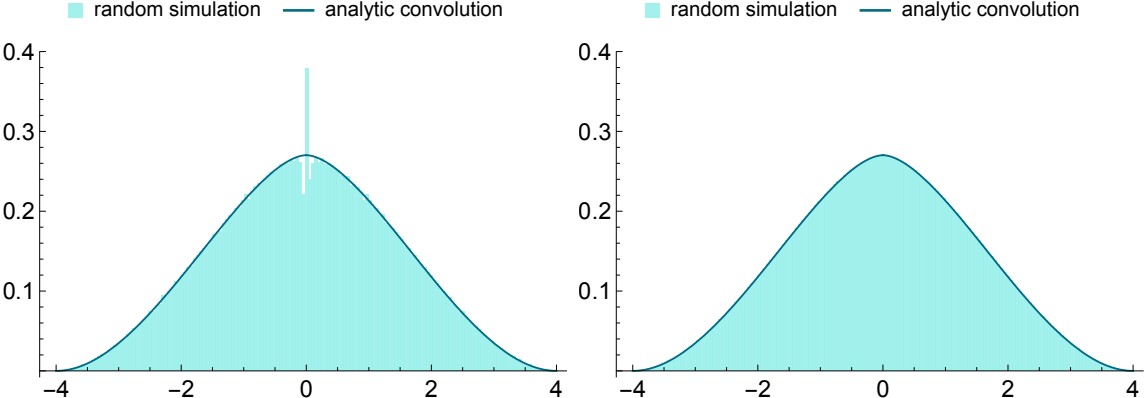

Figure 1: Comparison of GUE Liouvillian spectrums derived from Convolution of Hamiltonian spectrums versus Simulation. On the left, the simulation is obtained for $N = 60$ over 50 ensembles, while on the right, the simulation is generated for $N = 600$ over 50 ensembles, highlighting the degeneracy contribution diminishing by a factor of $1/N$.

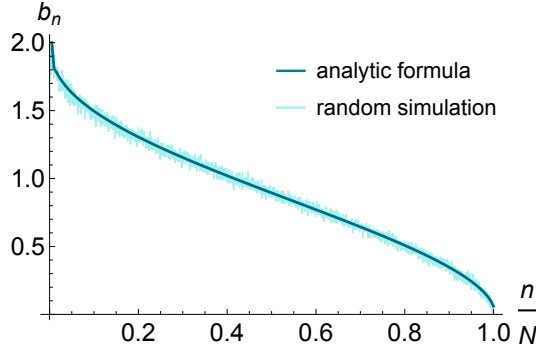

Figure 2: GUE Lanczos coefficients from Liouvillian spectrum.

## 5.3 Non-Gaussian ensemble

The Gaussian measure can be modified by substituting the Gaussian potential $H^2$ in $\mathrm{Tr}(H^2)$ with a generalized potential $V(H)$. A detailed elucidation of this method is presented in Appendix (B). Following [18], we explore extended theories featuring sextic and quartic potentials ($V_s$ and $V_q$ respectively) defined respectively by

$$V_s(E) = 3E^2 - E^4 + \frac{2}{15}E^6, \qquad V_q(E) = \frac{8}{3}E - \frac{4}{9}E^3 + \frac{1}{6}E^4. \tag{36}$$

Those potentials are associated with non-Gaussian Hamiltonian densities

$$\rho_s(E) = \frac{4 - E^2}{\pi}\left(\frac{7}{10} - \frac{3}{5}E^2 + \frac{1}{5}E^4\right), \quad \rho_q(E) = \frac{4 - E^2}{\pi}\left(\frac{1}{3} - \frac{1}{3}E + \frac{1}{6}E^2\right). \tag{37}$$

Following the generation of Gaussian matrices, our objective is to approximate the spectrum of the non-Gaussian Hamiltonian by stretching the Gaussian spectrum through convolution with the desired density of states [69]. This can be accomplished by initially calculating the cumulative distribution function of the Gaussian spectrum and subsequently determining the inverse cumulative distribution function of the target non-Gaussian distribution, denoted as $E_i' = g^{-1}(f(E_i))$. The

cumulative distribution functions for the Gaussian Unitary Ensemble (GUE) and the non-Gaussian target respectively are as follows,

$$f(E) = \int_{-2}^{E} dE' \frac{\sqrt{4 - E'^2}}{2\pi}, \qquad g(E) = \int_{-\infty}^{E} dE' \rho_{\text{non-Gauss.}}(E').$$ (38)

The unitary matrix diagonalizing the Hamiltonian always follows Haar statistics, allowing us to utilize the same matrix of eigenvectors to transform the Hamiltonian back to its original form, albeit with eigenvalues stretched accordingly. We deduce the respective Liouvillian spectrum and Lanczos coefficients, setting $N = 600$ and averaging over 50 ensembles. This is achieved through both analytical techniques and random simulations, as depicted in Fig. (3) and Fig. (4). In both cases, we find a perfect match between numerical and analytical approaches.

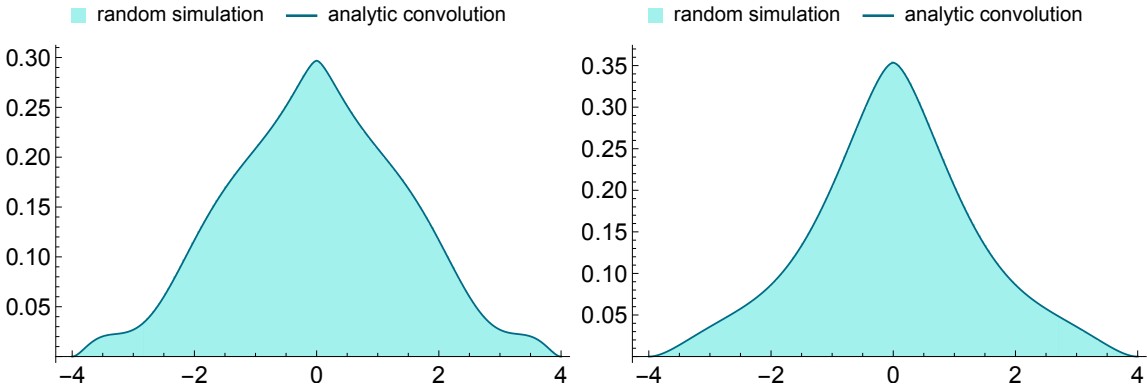

Figure 3: Non-Gaussian Liouvillian spectrum from Hamiltonian spectrum for sextic potential $V_s$ (left) and quartic potential $V_q$ (right) at $N = 600$ over 50 ensembles.

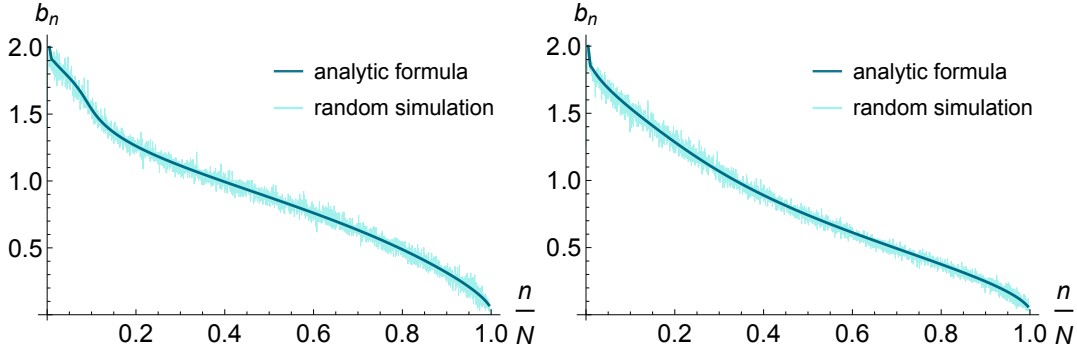

Figure 4: Non-Gaussian Lanczos coefficients from Liouvillian spectrum for sextic potential $V_s$ (left) and quartic potential $V_q$ (right).

## 5.4 SYK model

Random matrices provide effective approximations for characterizing the fine-grained spectrum of chaotic systems such as SYK. In particular, the quantum chaotic nature of the SYK model is validated by demonstrating that, over extended timescales approximating the Heisenberg time, the level statistics closely align with the predictions of random matrix theory, see [74]. In [75, 76], the

spectral density for the SYK model was derived using Q-Hermite polynomials. The detailed formula is described in Appendix (C). This section extends the derivation to deduce the spectral density of operators from such density of states. Subsequently, we employ the algorithm to verify the Lanczos spectrum for operators.

We consider the SYK model with $N = 14$ Majorana fermions strongly interacting in 4-body interaction, see Appendix (C). The Liouvillian spectrum derived from convolution versus simulation, and their corresponding Lanczos spectrum, are illustrated in Fig. (5).

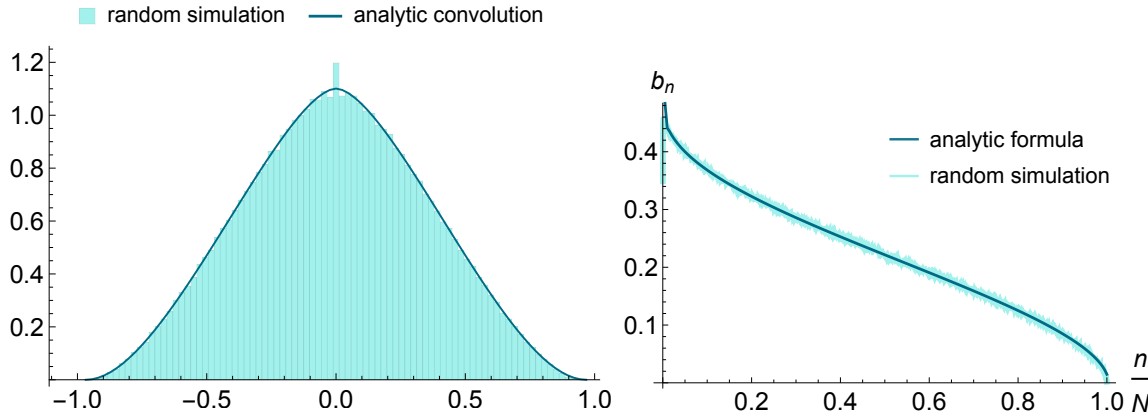

Figure 5: SYK $N = 14$ Liouvillian spectrum and Lanczos coefficients over 30 ensembles.

## 6   Discussion and Outlook

In this work, we have obtained the Lanczos spectrum of the Liouvillian superoperator associated with particular initial states. This has been achieved by leveraging knowledge of the Hamiltonian spectrum of the theory, particularly the density of states. We successfully verified the analytical predictions by exact numerical computations in several models of interest, such as interacting matrix models and SYK. It is quite remarkable that the Lanczos descent can be found in all cases using these techniques. Although we have not properly computed, this finding implies that spread complexity will saturate around the expected exponentially large time scale and around the expected exponentially large value.

There are several directions of interest for future developments. First, it would be desirable to have analytical formulas for different initial states. This would not affect too much the bulk of the spectrum we derived here but it will drastically change the behavior near the edge, for example, displaying the linear growth described in [1]. This problem already appears in the state evolution scenario [77]. Second, it would be interesting to find the statistics of the operator Lanczos spectrum, namely the two-point function. While the average Lanczos coefficients cannot discern between integrability and chaos, the variance of the Lanczos coefficients has been recently conjectured to be well approximated in Random Matrix Theory in [69]. It is then important to verify this conjecture for Heisenberg evolution as well. Another direction involves considering open systems, where the operator dynamics are dictated by a Lindbladian. The Lindbladian is composed of the Liouvillian, which accounts for the internal (intrinsic) operator dynamics, supplemented by an interaction term that represents the system's interactions with its environment. Aspects of the Krylov method in open systems were studied before [26,33,48,67,68,78] and we expect our methods will complement those. Finally, it would be very interesting to continue the study of the gravitational counterpart

of spread complexity. Recently, it has been shown this quantity can reproduce the volume of black hole interiors (Einstein-Rosen bridges), see [44, 53, 79] for the JT gravity scenario, and [77] for the case of general relativity in general dimensions. In this vein it would be interesting to derive the complexity slope of [18] using gravitational dynamics only.

## Acknowledgement

The author thanks Javier Magan for contributing ideas and guidance during the completion of this work, and for a careful reading and comments on this manuscript.

## A    Deriving Lanczos coefficients from densities of operators

Deriving Lanczos coefficients from Liouvillian densities can follow two methods akin to those used in deriving them from state densities, as discussed in [18]. The approaches are predicated on the assumption that the integration support for Eq. (15) shrinks monotonically as $x$ increases, as first noticed in [80, 81], and that $b(x)$ decrease monotonically, e.g. $b'(x) < 0$. We now define the density function of $b$ as $\gamma(b(x)) \equiv b'(x)$. Then

$$\rho(E) = \int_{b_0}^{0} -\gamma^{-1}(b(x)) \frac{H(4b(x)^2 - E^2)}{\pi\sqrt{4b(x)^2 - E^2}} \mathrm{d}b. \tag{39}$$

The previous monotonic assumptions allow us to represent $E$ and $b$ as decaying functions $b = b_0 e^{-z}$ and $E = E_0 e^{-\epsilon}$, so that

$$\rho\left(E_0 e^{-\epsilon}\right) = \int_0^\infty dz b_0 \gamma\left(b_0 e^{-z}\right) \frac{H\left(4 - E_0^2 b_0^{-2} e^{2(z-\epsilon)}\right)}{\pi b_0 \sqrt{4 - E_0^2 b_0^{-2} e^{2(z-\epsilon)}}}. \tag{40}$$

The above equation has the form

$$f(\epsilon) = \int_0^\infty dz g(z) h(\epsilon - z). \tag{41}$$

Deconvolving the above equation results in

$$g(z) = b_0 \gamma\left(b_0 e^{-z}\right). \tag{42}$$

From here, solving for $b'(x) = \gamma(b(x))$ provides us with $b(x)$.

Another way to solve for Lanczos coefficient $b(x)$ is to consider general cases (applicable for Lanczos coefficient $a(x) \neq 0$, see [18]). Here, we utilize this method only in the case $a(x) = 0$ for operators. First, the support of integrant (15), $E_{\text{left}} = -2b(x), E_{\text{right}} = 2b(x)$, is assumed to shrink as x increases. The cumulative densities of operators of Eq. (15) read

$$P(E) = \int_{E_{\min}}^{E} dE \rho(E) \approx \int_0^1 dx \int_{E_{\text{left}}(x)}^{E} dE \frac{H\left(4b(x)^2 - E^2\right)}{\pi\sqrt{4b(x)^2 - E^2}} \tag{43}$$

$$= \int_0^1 dx P_c\left(4 \frac{E - E_{\text{left}}(x)}{E_{\text{right}}(x) - E_{\text{left}}(x)} - 2\right), \tag{44}$$

with $P_c \equiv \int_{-2}^{z} \frac{1}{\pi\sqrt{4-z'^2}} dz'$ and cumulative distribution function $P(E) = \int_{E_{\min}}^{E} dE \rho(E)$. Notice that to arrive at the second line we used the substitution $E = bz'$.

The lower bound of the integral over $E$ in the first equation is set at $E_{\text{left}}(x)$ because a specific x only influences the density of operators for $E > E_{\text{left}}(x)$. Considering an $E$ equal to $E_{\text{left}}(X)$ for a certain $X$, our assumption of monotonicity implies that the density of operators for any $E < E_{\text{left}}(X)$ receives contributions solely from $x \leq X$. Consequently, the cumulative density of operators from $E = E_{\text{min}}$ to $E = E_{\text{left}}(X)$ is exclusively affected by $x \leq X$. Therefore, we can narrow the integration limits in Eq. (44) accordingly,

$$P(E_{\text{left}}(X)) = \int_0^X \mathrm{d}x P_c \left( 4 \frac{E_{\text{left}}(X) - E_{\text{left}}(x)}{E_{\text{right}}(x) - E_{\text{left}}(x)} - 2 \right), \tag{45}$$

$$1 - P(E_{\text{right}}(X)) = \int_0^X \mathrm{d}x \left[ 1 - P_c \left( 4 \frac{E_{\text{right}}(X) - E_{\text{left}}(x)}{E_{\text{right}}(x) - E_{\text{left}}(x)} - 2 \right) \right]. \tag{46}$$

In the second equation, we proceeded with the same steps, integrating from the opposite direction, to derive a corresponding equation for $E_{\text{right}}(X)$.

As these equations retrospectively reference the variable $X$, we can iteratively resolve this system for $E_{\text{left}}(x)$ and $E_{\text{right}}(x)$ as described in Algorithm (1), using the known functions $P(E)$ (obtainable from the density of operators) and $P_c(z)$ (computable via the defining integral). The Lanczos coefficients $b(x)$ can then be straightforwardly deduced by inverting the definitions of $E_{\text{left/right}}(x)$, namely $b(x) = (E_{\text{right}}(x) - E_{\text{left}}(x))/4$.

---

**Algorithm 1** Approximating solutions to the integral equation applicable for non-vanishing $a(x)$ (algorithm taken from [18])

---

1: $E_{\text{right}}(0) \leftarrow E_{\text{max}}$
2: $E_{\text{left}}(0) \leftarrow E_{\text{min}}$
3: **for** $m = 1$ to $M$ **do**
4:     Set $E_{\text{left}}\left(\frac{m}{M}\right)$ to be the lowest solution $E > E_{\text{left}}\left(\frac{m-1}{M}\right)$

$$P(E) = \frac{1}{M} \sum_{i=0}^{m-1} P_c \left( \frac{4(E - E_{\text{left}}\left(\frac{i}{M}\right))}{E_{\text{right}}\left(\frac{i}{M}\right) - E_{\text{left}}\left(\frac{i}{M}\right)} - 2 \right) \tag{47}$$

5:     Set $E_{\text{right}}\left(\frac{m}{M}\right)$ to be the highest solution $E < E_{\text{right}}\left(\frac{m-1}{M}\right)$

$$1 - P(E) = \frac{1}{M} \sum_{i=0}^{m-1} \left[ 1 - P_c \left( \frac{4(E - E_{\text{left}}\left(\frac{i}{M}\right))}{E_{\text{right}}\left(\frac{i}{M}\right) - E_{\text{left}}\left(\frac{i}{M}\right)} - 2 \right) \right] \tag{48}$$

6: **end for**
7: $a \leftarrow \frac{E_{\text{left}} + E_{\text{right}}}{2}$ (which vanishes for operators)
8: $b \leftarrow \frac{E_{\text{right}} - E_{\text{left}}}{4}$

---

## B Non-gaussian ensemble from GUE

Hermiticity allows a matrix M in Gaussian ensembles to be represented in diagonalized form as $M = U\Lambda U^\dagger$, where $\lambda = \text{diag}(\lambda_1, .., \lambda_N) \in \mathbb{R}$ is a diagonal matrix of N distinct real eigenvalues, and rows of the unitary matrix $U$ are their corresponding eigenvectors. The diagonalization is not unique due to the ordering of eigenvalues, there are $n!$ distinct diagonalizations in general, and U can vary as $U \to U \, \text{diag}(e^{i\phi_1}, ..., e^{i\phi_N})$ up to for any choice of phase $\phi_1, ..., \phi_N$. Additionally, the GUE

(GOE, GSE) is invariant under unitary (orthogonal, symplectic) conjugation $dM = d(VMV^{-1})$. In general, the Lebesgue measure for all ensembles of squared matrices is invariant under conjugation change of bases. It can be written as the product of the Lebesgue measures of all real components of the matrix,

$$dM = \prod_{i<j} dM_{i,i} \prod_{\alpha=0}^{\beta-1} dM_{i,j}^{(\alpha)} \,. \tag{49}$$

Under diagonalization $M \to (\Lambda, U)$, the corresponding Lebesgue measure can be factorized into $dM = |\Delta(\Lambda)|^\beta d\Lambda dU_{\text{Haar}}$, where $d\Lambda$ denotes the Lebesgue measure $d\Lambda = \prod_{i=1}^N d\lambda_i$, on $\Lambda$, and $dU_{\text{Haar}}$ is the canonical Haar measure on U, with the Vandemomonde determinant

$$\Delta(\Lambda) = \prod_{1 \le i < j \le N} (\lambda_j - \lambda_i) \,. \tag{50}$$

We can build measures from $dM$ by multiplication with the potential function $V(M)$ which is a polynomial of $Md\mu(M) = e^{-\operatorname{Tr} V(M)} dM$. The joint probability distribution density on eigenvalues is given by

$$P_{N\beta}(x_1, ..., x_N) = \frac{1}{Z_{N\beta}} e^{-\frac{\beta N}{4} \operatorname{Tr} V(\Lambda)} |\Delta(\Lambda)|^\beta \,, \tag{51}$$

and the constant $Z_{N\beta}$ is chosen such that the $P_{N\beta}$ is normalized to unity (partition function)

$$Z_{N\beta} \sim \int_{\mathbb{R}^n} d\Lambda |\Delta(\Lambda)|^\beta e^{-\frac{\beta N}{4} \operatorname{Tr} V(\Lambda)} \,. \tag{52}$$

Employing the Coulomb Gas technique, as detailed in [82], the principal value integral

$$\frac{1}{4} V'(\omega) = \text{p.v.} \int dE \frac{\rho(E)}{\omega - E} \tag{53}$$

results in the corresponding potential delineated in Eq. (36).

## C  SYK model analytic spectrum

The Sachdev-Ye-Kitaev model [71, 83] is a quantum mechanical system in $0+1$ dimension of $N$ fermions. The Hamiltonian engages those fermions in an all-to-all random quartic interaction. The model is notable for being solvable at strong coupling, exhibiting maximal chaos, and demonstrating emergent conformal symmetry [84–86]. The SYK Hamiltonian is given by [71] as

$$H = \frac{1}{4!} \sum_{i,j,k,l=1}^N J_{ijkl} \psi_i \psi_j \psi_k \psi_l \,, \tag{54}$$

where the Hermitian operators $\psi_i$ represent Majorana fermions that obey the Euclidean N-dimensional Clifford algebra $\{\psi_i, \psi_j\} = \delta_{ij}$. The coupling constants $J_{ijkl}$ are Gaussian random variables obeying the distribution

$$P(J_{ijkl}) = \sqrt{\frac{N^{q-1}}{2(q-1)!\pi J^2}} \exp\left(-\frac{N^{q-1} J_{ijkl}^2}{2(q-1)! J^2}\right) \,, \tag{55}$$

with zero disorder average and the variance given by [75] as

$$\overline{J_{ijkl}^2} = \binom{N}{q} \frac{J^2(q-1)!}{2^q N^{q-1}} . \tag{56}$$

The analytic expression for the SYK spectral density is well-approximated by Q-Hermite polynomials [76], with $Q = \eta_{N,q}$, as

$$\rho_{\mathrm{QH}}(E) = c_N \sqrt{1 - (E/E_0)^2} \prod_{k=1}^{\infty} \left[ 1 - 4\frac{E^2}{E_0^2} \left( \frac{1}{2 + \eta_{N,q}^k + \eta_{N,q}^{-k}} \right) \right] , \tag{57}$$

where $c_N$ is constant for normalization over $2^{N/2}$ states, we have defined $E_0^2 = (4\overline{J_{ijkl}^2})/(1 - \eta_{N,q})$, and $\eta_{N,d}$ is the suppression factor from commuting the products of 4 gamma matrices,

$$\eta_{N,q} = \binom{N}{q}^{-1} \sum_{r=0}^{q} (-1)^r \binom{q}{r} \binom{N-q}{q-r} . \tag{58}$$

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
