# Peer review of "Lanczos spectrum for random operator growth"

_SciPost Physics_

## Round 2 · Referee Report · Zhuo-Yu Xian (Referee 2) · 2024-5-12

Weaknesses
- This paper does not present significant novel contributions to the field.
- The author neglects the important spectral correlation in a random matrix.
- The initial operator in the Lanczos algorithm is unclear.
Report
In this paper, the author studies the spectral densities and the Lanczos coefficients of the Liouvillians given by random matrices. The Lanczos algorithm was nicely investigated in random matrix theories in previous literature. The author mainly borrows the techniques from https://arxiv.org/abs/2208.08452 and applies them to the Liouvillian acting on an operator instead of a state. However, the findings presented in this paper lack depth and do not sufficiently explore the complexities of the topic when compared to previous literature. For example, the author has not studied the variance of Lanczos coefficients, the Krylov wavefunction, nor the Krylov complexity, which were detailed in previous literature.
Besides that, this paper has some weaknesses in validity and clarity.
First, the author neglects the spectral correlation in Eq. (24). In RMT, the energy levels are actually correlated. The two-point function of spectral density in ensemble average is not factorizable. For example, in Gaussian ensembles, the spectral correlations are given by sine kernels.
Second, the author has not specified the initial operator in the Lanczos algorithm. The Lanczos coefficients not only depend on the spectrum of the Liouvillian but also on the initial operator. Since the author directly adopts the method from https://arxiv.org/abs/2208.08452, I think that the initial operator may be a random operator generated by the Haar random unitary in the double-copy Hilbert space, namely, the space of the index pair $(i,j)$.
Considering the lack of novelty, validity, and clarity in this paper, I do not recommend it for publication in SciPost. I will recommend it to be transferred to another journal if the author can address the above two points regarding validity and clarity.
Besides that, this paper has some weaknesses in validity and clarity.
First, the author neglects the spectral correlation in Eq. (24). In RMT, the energy levels are actually correlated. The two-point function of spectral density in ensemble average is not factorizable. For example, in Gaussian ensembles, the spectral correlations are given by sine kernels.
Second, the author has not specified the initial operator in the Lanczos algorithm. The Lanczos coefficients not only depend on the spectrum of the Liouvillian but also on the initial operator. Since the author directly adopts the method from https://arxiv.org/abs/2208.08452, I think that the initial operator may be a random operator generated by the Haar random unitary in the double-copy Hilbert space, namely, the space of the index pair $(i,j)$.
Considering the lack of novelty, validity, and clarity in this paper, I do not recommend it for publication in SciPost. I will recommend it to be transferred to another journal if the author can address the above two points regarding validity and clarity.
Recommendation
Accept in alternative Journal (see Report)
Weaknesses
(1) Does not add much to the field. Most of it already known.
Report
Dear Editor,
In this paper authors have studied Lanczos algorithm and spectrum for Random matrix theory (RMT). Previously it has been studied for RMT in state space and complexity associated with spread of state has been computed for Gaussian, Non-Gaussian as well as RMT with noise in various papers (e.g. https://arxiv.org/abs/2208.08452, https://arxiv.org/abs/2303.12151, https://arxiv.org/pdf/2307.15495). In fact, in http://arxiv.org/abs/2303.12151, it has been investigated how the nature of the complexity is dictated by the density of states.
In the paper, authors studied the Lanczos algorithm for Liouvillian. Only difference between this paper and previous papers is that, unlike the previous papers where the focus is on the space of state , here the author is focusing on the space of operators. As well know in the space of operators, the diagonal elements of Krylov space (a_n's) vanishes and one only gets the b_n's which are same as those found in https://arxiv.org/abs/2208.08452. Technique wise there is nothing much new in this paper. Computations of https://arxiv.org/abs/2208.08452 has been adapted straightforwardly for the same models of RMTs here and expectedly author obtained same b_n 's and there are no a_n's.
I do not think this study add significantly to whatever has been done already, although it is correct. Hence I will not be able to recommend this for publication in SciPost, given its high criterion. I will recommend it to transfer to other journal unless the author significantly add newer results which are not discussed in those above mentioned papers and bring out the usefulness of studying this for RMT in operator space.
Best,
Referee
In this paper authors have studied Lanczos algorithm and spectrum for Random matrix theory (RMT). Previously it has been studied for RMT in state space and complexity associated with spread of state has been computed for Gaussian, Non-Gaussian as well as RMT with noise in various papers (e.g. https://arxiv.org/abs/2208.08452, https://arxiv.org/abs/2303.12151, https://arxiv.org/pdf/2307.15495). In fact, in http://arxiv.org/abs/2303.12151, it has been investigated how the nature of the complexity is dictated by the density of states.
In the paper, authors studied the Lanczos algorithm for Liouvillian. Only difference between this paper and previous papers is that, unlike the previous papers where the focus is on the space of state , here the author is focusing on the space of operators. As well know in the space of operators, the diagonal elements of Krylov space (a_n's) vanishes and one only gets the b_n's which are same as those found in https://arxiv.org/abs/2208.08452. Technique wise there is nothing much new in this paper. Computations of https://arxiv.org/abs/2208.08452 has been adapted straightforwardly for the same models of RMTs here and expectedly author obtained same b_n 's and there are no a_n's.
I do not think this study add significantly to whatever has been done already, although it is correct. Hence I will not be able to recommend this for publication in SciPost, given its high criterion. I will recommend it to transfer to other journal unless the author significantly add newer results which are not discussed in those above mentioned papers and bring out the usefulness of studying this for RMT in operator space.
Best,
Referee
Recommendation
Accept in alternative Journal (see Report)

---

## Editorial Decision

awaiting_resubmission